

# Flood forecasting using transboundary data with the fuzzy inference system: The Maritza (Meriç) River

Abdurrahim Aydın[1], İbrahim Yücedağ[2], Remzi Eker[1]

[1]Düzce University, Faculty of Forestry, 81620 Düzce, Turkey
[2]Düzce University, Faculty of Technology, 81620 Düzce, Turkey

*Correspondence to:* Remzi Eker (remzieker@duzce.edu.tr)

**Abstract.** The rising floodwaters of the Maritza River, originating in the Balkans, affect those living in adjacent areas. The lower regions of the Maritza River are especially vulnerable to floods. The city of Edirne and the surrounding region in Turkey located downstream of the Maritza River are frequently affected by flooding. The section of the river inside the Turkish border is short; therefore, there is not adequate warning time for Turkey to alert the population against flash floods. For this reason, it is essential that Turkey acquire current flow information from Bulgarian sites for use in current flow prediction for Turkey's section of the river. To this aim, in order to predict the current flow of the Kirişhane station (Turkey) from the transboundary data of Plovdiv and Svilengrad stations (Bulgaria), four different models ($M_1$–$M_4$) were developed by using the fuzzy inference system (FIS). Flow data from the Plovdiv, Svilengrad and Kirişhane stations were gauged every two hours covering the period from 9 February 2010 00:00:00 to 21 February 2010 22:00:00. In the first model, estimation was made using the current flows of the Plovdiv and Svilengrad stations. In the second model, estimation was made based on a two hour ahead prediction of the Svilengrad station and a four hour ahead prediction of the Plovdiv station. In the third model, calculations were based on predictions of four hours ahead of the Svilengrad station and eight hours ahead of the Plovdiv station. In the last model, estimation was based on predictions of six hours ahead of the Svilengrad station and twelve hours ahead of the Plovdiv station. The prediction ability of the FIS was evaluated by using the determination coefficient ($R^2$), the Nach-Sutcliffe sufficiency score (NSSS), the normalized root mean square error (NRMSE), and the correlation coefficient (CORR). According to their performance criteria, all developed models produced highly satisfactory results. The $M_2$ model gave the best results according to both NSSS and $R^2$ values, whereas $M_4$ had the lowest NSSS and $R^2$ values. This study demonstrated that developed fuzzy rule-based models can satisfactorily predict flood waves and thus can be used for flood forecasting and warning systems.

**Keywords:** FIS, flood, streamflow forecasting, transboundary data, Maritza River



# 1 Introduction

Within flow regulation and water resources management studies, streamflow forecasting is a very demanding task which seeks to mitigate the effects of floods on human and dam safety as well as on ecosystem sustainability (Lekkas et al., 2005). However, the process of forecasting streamflow is a very complex hydrological process owing to the tremendous spatial and temporal variability in the characteristics of terrain and rainfall patterns in conjunction with other variables associated with modeling (Tokar and Markus, 2000; Nayak et al., 2005). Reliable water level forecasts enable the use of early warning systems to alert the population as well as real time control of hydraulic structures in order to mitigate the adverse effects when floods occur (Alvisi et al., 2006). Recording and analyzing the streamflow are indispensable procedures because they can generate significant indications of both past and future flow characteristics (Küçük and Ağıralioğlu, 2006). Furthermore, flood management studies require knowledge of the magnitude and frequency of high flows (Amisigo et al., 2008). Accurate and timely prediction of high and low flow events can provide the information required to make strategic decisions at any watershed location (Besaw et al., 2010). Hence, the forecasting of stream flows in real time has received noticeable attention from hydrologists and resource engineers for many decades (Chang and Chen, 2001).

To date, a wide variety of models for streamflow forecasting have been developed and applied, ranging from completely black box models to very detailed conceptual models (Nayak et al., 2005). The methods used for forecasting gauged and/or ungauged streamflow are categorized as conceptual, metric, physics-based and data-driven (Besaw et al., 2010). Data-driven methods have been extensively adopted for forecasting streamflow. Of the data-driven methods, multiple linear regression (MLR), auto-regressive moving average (ARMA), and artificial neural networks (ANNs) are commonly used models (McKerchar and Delleur, 1974; Chaloulakou et al., 1999; Cığızoğlu, 2003; Yurekli et al., 2005; Fırat, 2008; Wang et al., 2008; Aydın and Eker, 2012) because they require much less development time and are capable of more accurate prediction (Govindaraju, 2000). In addition, fuzzy logic has been used in hydrological modeling for various applications (Altunkaynak et al., 2005; Nayak et al., 2005; Chen et al., 2006; Fırat and Güngör, 2007; Özger, 2009; Şen and Altunkaynak, 2009; Turan and Yurdusev, 2009). Fuzzy models enable nonlinear solutions in practice when the required number of fuzzy sets and rules is provided (Özger, 2009).

Floodwaters rising from the Maritza River, which is the biggest river on the Balkan Peninsula, affect Turkey, Greece, and Bulgaria. In particular, the lower regions of the Maritza River are vulnerable to floods because the physical and hydrological characteristics of the river create a high flooding potential. In Turkey, the city of Edirne and the surrounding region are prone to flooding. The flood of June 1911 was one of the biggest, with a maximum discharge of approximately 1500 m³/s (Tuncok, 2015). Some of the more recent floods documented include the ones in 1957 with 1200 m³/s and in 1976 with 783 m³/s in addition to those in 2005, 2006, and 2007 (Tuncok, 2015). Moreover, the flood occurring on 16 February 2010 is considered to be the second biggest flood in the last 26 years. During this flood, a maximum discharge of 1713 m³/s was measured in the



city of Edirne, while 2800 m³/s was measured in Ipsala (Batur and Maktav, 2012). Turkey is unable to provide adequate warning time to alert the population against floods because the section of the river inside the country is short (Sezen et al., 2007). Thus, getting current flow information from Bulgarian sites and using it in the current prediction for Turkey's section of the River is essential. The main aim of the present study was to develop a fuzzy model from flow data originating from three stations (Plovdiv and Svilengrad in Bulgaria and Kirişhane in Turkey) located on the Maritza River, and to show the capability of the developed models to use the transboundary data in the prediction of current streamflow.

## 2 Material and method

### 2.1 Fuzzy logic

Fuzzy logic, which was suggested by Zadeh in 1965, is a generalization of classical logic. The difference between fuzzy logic and crisp (i.e., classical) logic is established by introducing a membership function. A universal set X determined to be either members or non-members of a crisp set can be defined by a discrimination function and for a given crisp set A. This function, which can be indicated by $\mu_A : X \rightarrow \{0,1\}$, assigns the value $\mu_A(x)$ to every $x \in X$ as follows (Hong and Lee, 1996):

$$\mu_A(x) = \begin{cases} 1 \ \textit{if and only if } x \in A, \\ 0 \ \textit{if and only if } x \notin A, \end{cases} \tag{1}$$

The membership function in fuzzy logic is a generalized form of the discriminant function; values assigned to elements in fuzzy sets fall within a specified range and are referred to as the membership grades of these elements. Membership function can be indicated by

$$\mu_A : X \rightarrow [0,1] \tag{2}$$

where X refers to the universal set defined in a specific problem, and [0, l] denotes the interval of real numbers from 0 to 1, inclusively.

Although a great number of membership functions were developed in fuzzy logic, the most commonly used are the triangular, sigmoidal, trapezoidal, and Gaussian membership functions (Baykal and Beyan, 2004). The most extensively used primitives for fuzzy union and fuzzy intersection are max and min operators. Assuming that A and B are two fuzzy sets with membership functions of $\mu_A$ and $\mu_B$, the fuzzy union and fuzzy intersection are as follows:

$$\begin{aligned} \mu_{A \cup B} = \max\{\mu_A, \mu_B\} \\ \mu_{A \cap B} = \min\{\mu_A, \mu_B\} \end{aligned} \tag{3}$$

Fuzzy logic provides a very valuable flexibility for reasoning which makes it possible to take into account inaccuracies and uncertainties. Additionally, fuzzy logic provides for the processing of linguistic knowledge and its corresponding numerical data through membership functions (Zadeh, 1973). Fuzzy models are capable of incorporating knowledge from human experts





naturally and conveniently, while traditional models fail to do so (Yen and Langari, 1999). Moreover, fuzzy models have the ability to handle the nonlinearity and interpretability features of the models (Yen and Langari, 1999). Fuzzy models can be created from knowledge of experts by translating linguistic information into fuzzy rules (Yanar, 2010), albeit there is no standard method available for transforming this knowledge (Jang, 1993).

A basic fuzzy logic statement is made in a rule-based system and can appear in the form of IF–THEN statements. A fuzzy inference system (FIS) can have both fuzzy sets and crisp values, but the outputs are always fuzzy sets. Hence, defuzzification is performed to convert fuzzy sets of outputs to crisp values. The general structure of an FIS is depicted in Fig. 1.

## 2.2 Model validation

The prediction ability of FIS is evaluated by using the determination coefficient ($R^2$), the Nach-Sutcliffe sufficiency score

(NSSS), the normalized root mean square error (NRMSE), and the correlation coefficient (CORR). The calculation of $R^2$ is given in Eq. (4):

$$R^2 = \left[ \frac{\sum_{i=1}^{N}(Q_O(t)-\overline{Q_O})(Q_P(t)-\overline{Q_P})}{\sqrt{\sum_{i=1}^{N}(Q_O(t)-\overline{Q_O})^2(Q_P(t)-\overline{Q_P})^2}} \right]^2 \tag{4}$$

where $Q_O$ represents the observed flow values, $Q_P$ represents the predicted flow values, $\overline{Q_O}$ is the mean of the observed flow values and $\overline{Q_P}$ is the mean of the predicted flow values.

The NSSS is commonly used to assess the predictive power of hydrological discharge models. It is defined as:

$$NSSS = 1 - \frac{\sum_{i=1}^{N}(Q_O(t)-Q_P(t))^2}{\sum_{i=1}^{N}(Q_O(t)-\overline{Q_O})^2} \tag{5}$$

The NSSS can range from $-\infty$ to 1. An NSSS equal to 1 means there is a perfect match between the model and the observations. The NRMSE statistic indicates a model's ability to predict a value away from the mean. The NRMSE is calculated by Eq. (6):

$$NRMSE = \frac{\left[ \sum_{i=1}^{N} \frac{\left( Q_P(t)-Q_O(t) \right)^2}{N} \right]^{0.5}}{\frac{1}{N}\sum_{i=1}^{N} Q_O(t)} \times 100 \tag{6}$$

The correlation coefficient (CORR) is a commonly used statistic for providing information about the strength of the relationship between the observed and the predicted data. If this criterion equals zero, then the model represents a perfect fit, which is categorically impossible. The CORR is calculated as in Eq. (7):

$$CORR = \frac{\sum_{i=1}^{N}(Q_O(t)-\overline{Q_O})(Q_P(t)-\overline{Q_P})}{\sqrt{\sum_{i=1}^{N}(Q_O(t)-\overline{Q_O})^2(Q_P(t)-\overline{Q_P})^2}} \tag{7}$$



## 2.3 Data and model application

The Maritza River, which is 490 km long, originates at 2400 m a.s.l in the Rila Mountains (Bulgaria) and flows southeast for 320 km (Fig. 2). The catchment of the Maritza River has an area of 52600 km$^2$, 28% of which is located in Turkey. After flowing along a short portion of the Greek–Bulgarian border, the river arrives in Turkey, where it flows for 13 km, after which it forms the border between Turkey and Greece until it disembogues into the Aegean Sea (Yıldız et al., 2014). The Maritza basin is under the influence of both the continental and the Mediterranean climates. The high flow period for the regions under the continental climatic influence can be observed in the northern part of Maritza basin and comes in late spring. On the other hand, the Mediterranean climatic influence is more visible to the south, affecting the lower parts of the Maritza and causing high flow conditions during winter (Tuncok, 2015). Flooding of the Maritza River typically occurs in the autumn, winter, and spring seasons and is caused mainly by heavy rainfalls and snowmelts (Tuncok, 2015).

In the present study, flow data from the Plovdiv and Svilengrad stations (Bulgaria), and the Kirişhane station (Turkey), all located on the Maritza River, were gauged every two hours covering the period from 9 February 2010 00:00:00 to 21 February 2010 22:00:00. These data were used to predict current streamflow via FIS. Figure 3 shows the two hourly streamflow data including the 159 rows used in this study. An example of one day's worth of data is given in Table 1.

In the study, four different models were developed for the prediction of the current flow of the Kirişhane station ($Q\_K_i$) using the transboundary data of the Plovdiv and Svilengrad stations. To this aim, the FIS editor graphical user interface (GUI) toolbox in the MATLAB/Simulink program was used. In the first model, $Q\_K_i$ was estimated using the current flows of the Plovdiv ($Q\_P_i$) and the Svilengrad ($Q\_S_i$) stations. In the second model, $Q\_K_i$ was estimated based on a two hour ahead predicted flow of the Svilengrad station ($Q\_S_{i-2}$) and a four hour ahead flow prediction of the Plovdiv station ($Q\_P_{i-4}$). In the third model, $Q\_K_i$ was estimated based on a four hour ahead predicted flow of the Svilengrad station ($Q\_S_{i-4}$) and an eight hour ahead flow prediction of the Plovdiv station ($Q\_P_{i-8}$). In the last model, $Q\_K_i$ was estimated based on a six hour ahead predicted flow of the Svilengrad station ($Q\_S_{i-6}$) and a twelve hour ahead predicted flow of the Plovdiv station ($Q\_P_{i-12}$). All developed models are shown in Table 2.

All four models were based on the Mamdani model, in which both input and output variables are fuzzified. The triangular membership function was selected for the two inputs (flows of Plovdiv and Svilengrad stations) and one output (flow of Kirişhane station) of the models (Fig. 3). The ranges of inputs and output variables were divided into fuzzy regions which included $S_N$ (Small N),…, S1 (Small 1), CE (Center), $B_1$ (Big 1),…, $B_N$ (Big N). The number of membership functions were increased in all models until the best result was obtained. The number of membership functions used in all models was the same for the two inputs and one output. In total, 26 membership functions were defined. After determination of the degrees of the data in different regions in which numbers can be different, the fuzzy rules of the models were generated by using AND





logical conjunction (Fig. 4). The centroid defuzzification procedure was employed to obtain the predicted flow value of the output based on the fuzzy rule base. The performances of the developed models were then compared. Figure 5 depicts examples of fuzzy rules developed in the models.

## 3 Results and discussion

The FIS was used to predict the current flow of the Kirişhane station from the transboundary data of the Plovdiv and Svilengrad stations on the Maritza River. To this aim, four different models were developed. The results of the developed models graphically shown in Fig. 6 were compared in terms of prediction performance. The NSSS, $R^2$, NRMSE, and CORR were used to evaluate the prediction performance of the models. For comparison, charts resulting from observed and predicted values for all models can be seen in Fig. 7.

A summary of the results of the performance criteria of the models is given in Table 3. According to the performance criteria of the models, all the developed models produced very satisfactory results. Among the developed models, $M_2$ gave the best results according to both NSSS and $R^2$ values, whereas $M_4$ had the lowest NSSS and $R^2$ values. According to the NRMSE criteria, $M_2$ gave the best result. According to the CORR statistic, although all the developed models had similar values, a stronger relationship was observed between the predicted and observed values in $M_4$. In all developed models, the same number

of triangular membership functions (equal to 26) was used. Except for $M_3$, the number of fuzzy rules written in all models was the same as seen in Table 3.

## 4 Conclusions

The main objective of the present study was to employ FIS for the prediction of the current flow of the Kirişhane station from the transboundary data of the Plovdiv and Svilengrad stations on the Maritza River. For this task, four different models were

developed. In the first model, an estimation was made using current flows of the Plovdiv and Svilengrad stations. In the second model, an estimation was made based on a two hour ahead flow prediction of the Svilengrad station and a four hour ahead flow prediction of the Plovdiv station. In the third model, a calculation was based on a four hour ahead flow prediction of the Svilengrad station and an eight hour ahead flow prediction of the Plovdiv station. In the last model, an estimation was based on a six hour ahead flow prediction of the Svilengrad station and a twelve hour ahead flow prediction of the Plovdiv station.

The models with the best fit were selected based on the values of a series of performance measures that included the $R^2$, the NSSS, the NRMSE, and the CORR. The results for each of the models are shown in Table 3. Based on these values, it can be observed that all the models developed were capable of yielding satisfactory predictions. According to the performance criteria



of the developed models, $M_2$ had a better prediction performance than the others. Model $M_2$ predicted the current discharge of the Kirishane station two hours ahead of the Svilengrad station and four hours ahead of the Plovdiv station.

The Maritza River, which is the biggest river on the Balkan Peninsula, affects Turkey, Greece, and Bulgaria. In particular, the lower regions of the Maritza River have suffered from floods. In this regard, although early warning systems offer the

population time to evacuate before floods, improving early warning systems involves multiple components, each with a cost. Furthermore, long-term observation of the hydro-meteorological conditions of the floods is needed. Hence, developed models are crucial in the prediction of current flows. Doing nothing is not an option; thus, this study has shown that developed fuzzy rule-based models can satisfactorily predict the flow regime with high accuracy and therefore can be used to develop a reliable flow forecasting system. Moreover, in practice, such studies may be used to apply the values of upstream stations for the

estimation of the missing values of downstream stations.

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





**Table 1: Example of one day's data as used in the study**

| Date | Hours | Stream Flow ($m^3/s$) | | |
| --- | --- | --- | --- | --- |
| | | Plovdiv Station | Svilengrad Station | Kirişhane Station |
| 09.02.2010 | 00:00 | 78.7 | 176.0 | 620.0 |
| | 02:00 | 81.5 | 171.0 | 632.0 |
| | 04:00 | 67.5 | 166.0 | 632.0 |
| | 06:00 | 55.8 | 161.0 | 624.0 |
| | 08:00 | 51.1 | 155.0 | 603.0 |
| | 10:00 | 48.7 | 150.0 | 512.0 |
| | 12:00 | 48.7 | 145.0 | 485.0 |
| | 14:00 | 53.4 | 141.0 | 435.0 |
| | 16:00 | 65.2 | 136.0 | 399.0 |
| | 18:00 | 65.2 | 134.0 | 377.0 |
| | 20:00 | 65.2 | 134.0 | 388.0 |
| | 22:00 | 92.7 | 129.0 | 421.0 |



**Table 2: Developed models, inputs and output**

| Model Name | Inputs | Output |
|---|---|---|
| $M_1$ | $Q\_P_i$, $Q\_S_i$ | $Q\_K_i$ |
| $M_2$ | $Q\_P_{i-4}$, $Q\_S_{i-2}$ | $Q\_K_i$ |
| $M_3$ | $Q\_P_{i-8}$, $Q\_S_{i-4}$ | $Q\_K_i$ |
| $M_4$ | $Q\_P_{i-12}$, $Q\_S_{i-6}$ | $Q\_K_i$ |

20





**Table 3: Results of performance criteria of developed models**

| Models | NSSS | $R^2$ | NRMSE | CORR | Number of Rules |
|--------|------|-------|-------|------|-----------------|
| $M_1$ | 0.979 | 0.980 | 0.049 | 7.98 | 99 |
| $M_2$ | **0.986** | **0.990** | **0.038** | 7.84 | 99 |
| $M_3$ | 0.980 | 0.984 | 0.046 | 7.54 | 98 |
| $M_4$ | 0.950 | 0.970 | 0.067 | **7.23** | 99 |



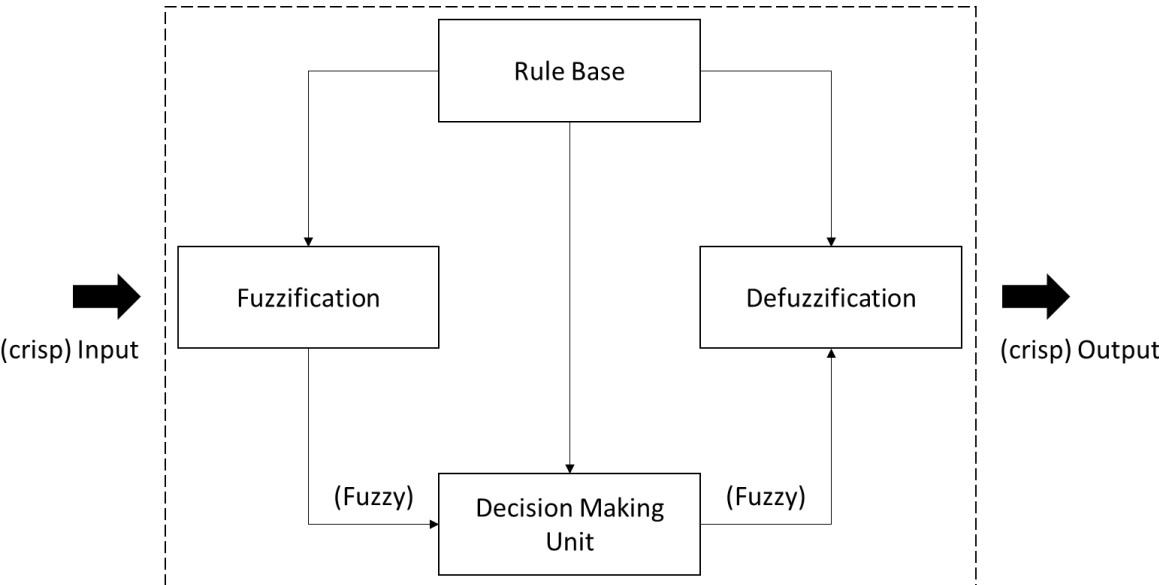

**Figure 1: Fuzzy inference system (FIS)**




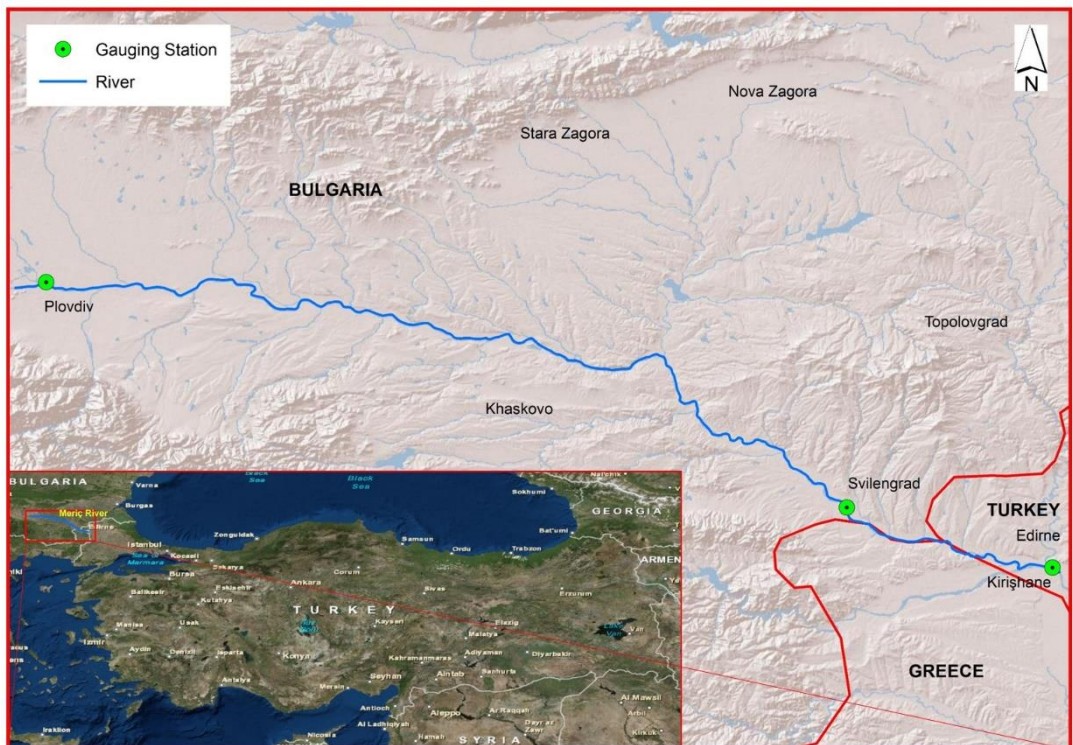

**Figure 2: Location map of Maritza River**





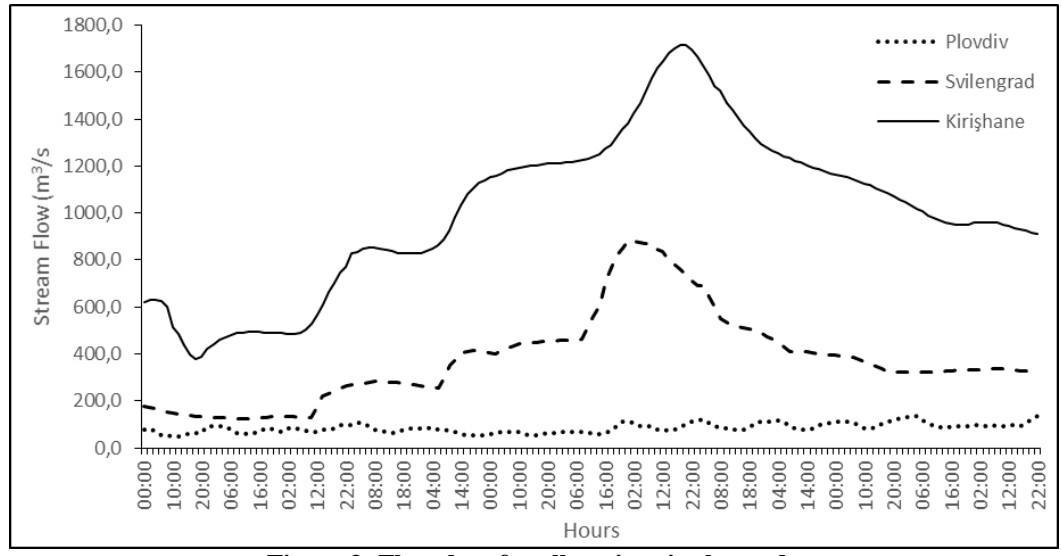

**Figure 3: Flow data for all stations in the study**





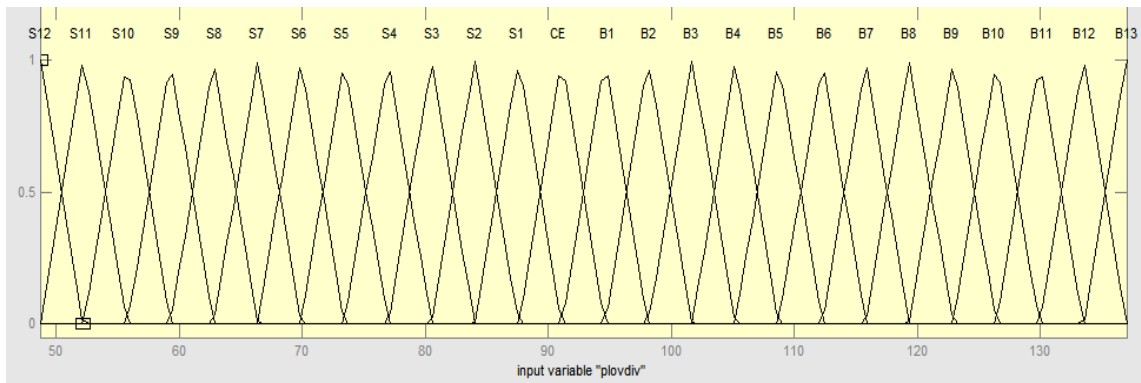

**Figure 4: Example depiction of triangular membership functions of flow data (Plovdiv station)**



1. If (plovdiv is CE) and (svilengrad is S12) then (kirishane is S12) (0.1)
2. If (plovdiv is CE) and (svilengrad is B12) then (kirishane is B9) (1)
3. If (plovdiv is CE) and (svilengrad is B3) then (kirishane is B9) (1)
4. If (plovdiv is CE) and (svilengrad is S3) then (kirishane is B4) (1)
5. If (plovdiv is CE) and (svilengrad is S5) then (kirishane is S1) (1)
6. If (plovdiv is CE) and (svilengrad is S6) then (kirishane is S1) (1)
7. If (plovdiv is S9) and (svilengrad is S12) then (kirishane is S9) (1)
8. If (plovdiv is S9) and (svilengrad is S7) then (kirishane is S12) (0.1)
9. If (plovdiv is S9) and (svilengrad is S3) then (kirishane is CE) (1)
10. If (plovdiv is S9) and (svilengrad is S2) then (kirishane is B5) (1)
11. If (plovdiv is S9) and (svilengrad is B2) then (kirishane is B13) (1)
12. If (plovdiv is S8) and (svilengrad is S12) then (kirishane is S12) (0.5)
13. If (plovdiv is S8) and (svilengrad is S3) then (kirishane is B2) (1)
14. If (plovdiv is S8) and (svilengrad is CE) then (kirishane is B4) (1)

**Figure 5: Example depiction of fuzzy rules developed in models**





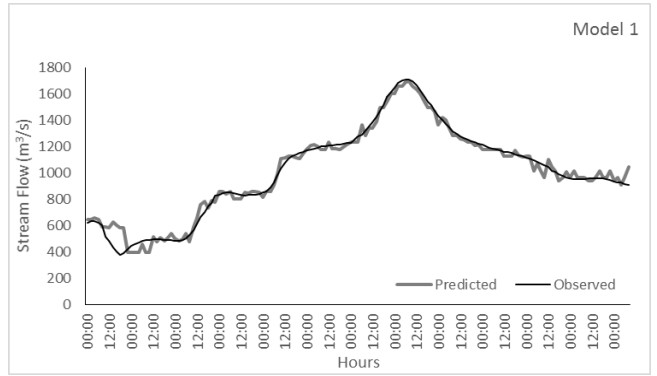
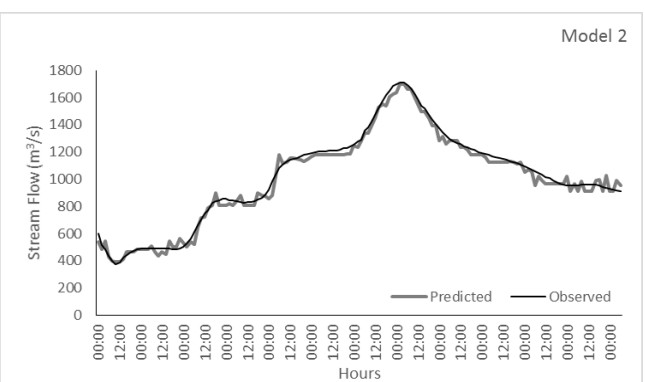

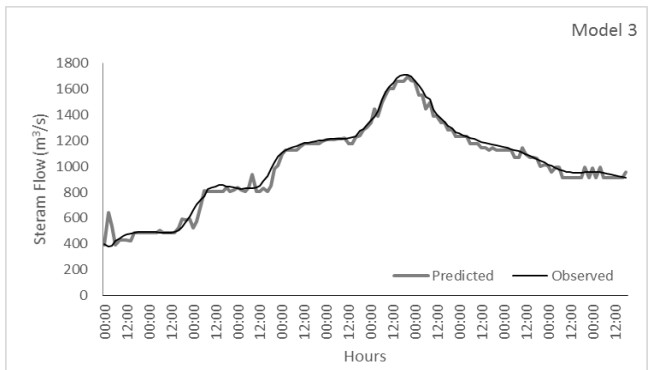
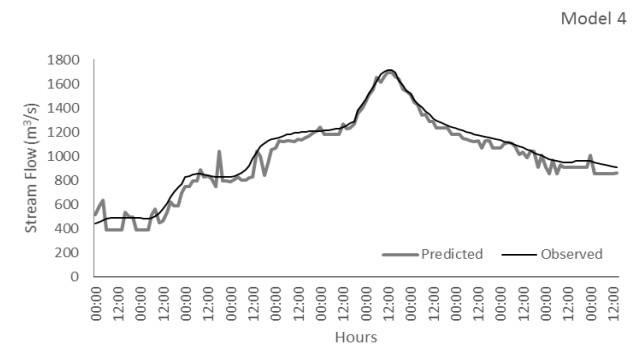

**Figure 6: Graphical depiction of the results of Models 1 – 4**




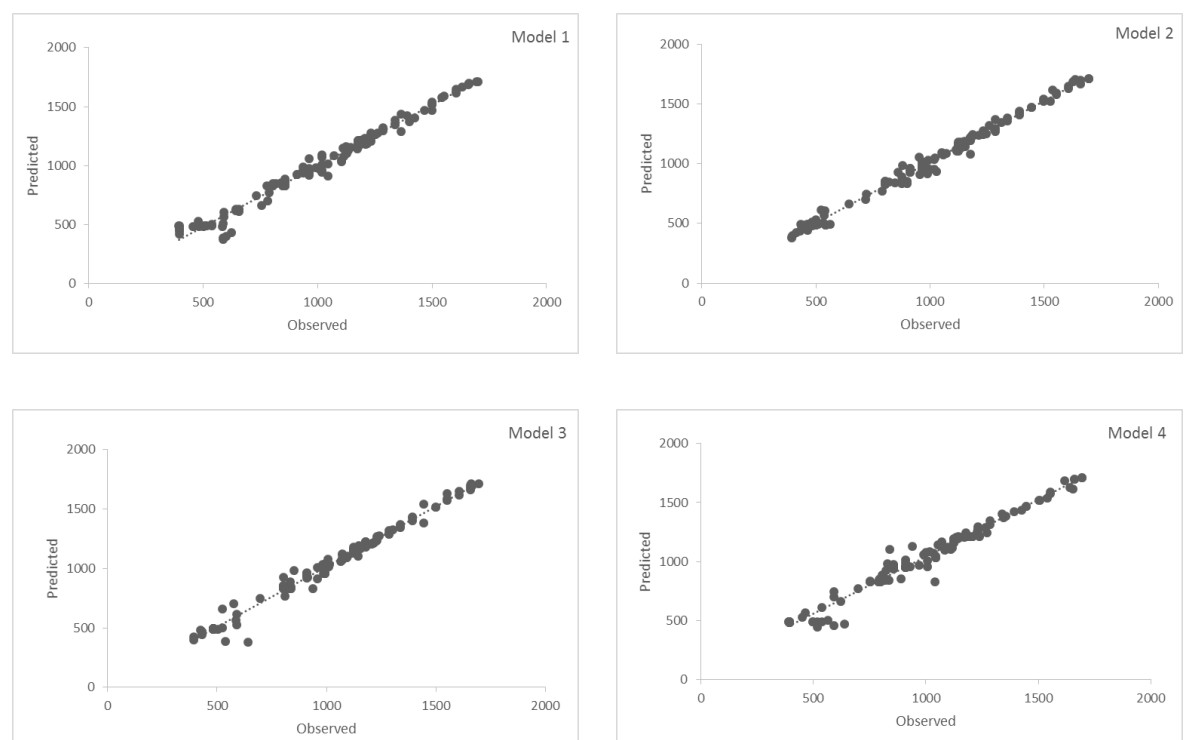

**Figure 7: Resulting scatterplots of observed and predicted values of developed models**