# Peer review of "Flood forecasting using transboundary data with the fuzzy inference system: The Maritza (Meriç) River"

_Natural Hazards and Earth System Sciences, 2016_

## Referee Comment (RC1) · Anonymous Referee #1 · 8 Jun 2016

This paper presents the application of the fuzzy inference system (FIS) to predict the streamflow at the Kirishane gauging station located in the River Maritza. The input data is the streamflow recorded at two gauging stations located upstream the site of interest: Plovdiv and Svilengrad stations. Four models are tested with different lag times. All of them use a triangular fuzzifier function with 26 members. The results show that the best model is the M2 that uses the predicted flow at Svilengrad in 2 hours and at Plovdiv in 4 hours as input. The authors conclude that the model can predict floods satisfactorily.

However, the paper in its current form has some important drawbacks that should be overcome before it can be considered for publication:

- The observed data series used in the paper is quite short. A series of 12 days with a temporal resolution of two hours that includes an only flood event is used to calibrate the model. A longer streamflow series is required to calibrate and validate the proposed model. On one hand, a longer streamflow series should be used to calibrate the model, including a variety of flood events and catchment responses. In the other hand, some flood events should be used to validate the model.

- Another essential task is to show how the model improves other existing or simpler forecast models. Maybe, a simpler deterministic model that uses the travel time through the reach could give similar results.

- The model has not been validated. This is an essential step that should be included in the paper. Section 2.2 is not the model validation, but the model selection.

- The fuzzy inference system and the Mamdani model should be described in more detail. A section should be devoted to describe the proposed model.

- Different fuzzifier functions should be proved. The results of the sensitivity analysis to select the number of membership functions should be included in the paper.

- Something seems to be wrong with the determination coefficient formula shown in Eq. 4.

- The correlation coefficient values shown in Table 3 seem to be wrong. The correlation coefficient should give values between -1 and 1.

- Table 1 and Figure 5 should be removed from the paper.

- The paper concludes that the best model uses the prediction of streamflow at Svilengrad two hours ahead and at Plovdiv four hours ahead. I am not sure the authors could have these predictions to use the proposed model in real time. In addition, the Kirishane station is downstream the Svilengrad and Plovdiv stations. Consequently, it should be more coherent to use recorded streamflow in the past to forecast the streamflow at Kirishane in the future.

---

## Referee Comment (RC2) · Anonymous Referee #2 · 3 Aug 2016

In this study the authors present, in a very synthesized way, the results of the application of a data-based method (the Fuzzy Inference System) for flow forecasting in the River Maritza at the Turkish city of Kirişhane. I understand that the model developed by the authors might have a relevance in the study region, but in order to fulfill the standards of a scientific publication, a considerable amount of additional work needs to be done by the authors.

The main weaknesses of the study are, in order of importance:

(1) The authors do not validate the forecast model. There are many strategies for validation, but probably the most straightforward one is the split-sample validation (see e.g. Klemes 1986). In this study, the authors use the same period (9-21 February 2010)

[Figure]

for the calibration and for the validation of the model. Reporting the performance measures of the model(s) for this periods gives only a goodness-of-fit measure. Presenting them as validation measures (section 2.2 Model validation) is scientifically wrong, specially for a data-based method. Data-based methods are particularly characterized by performing well inside the range of values for calibration, and performing more poorly, even catastrophically outside these ranges. These connects another issue, related with the period used. Why just 9-21 February? Is this flood event representative in terms of processes [see also comment (6) below]? Is the reason for the short period chosen related with data availability/limitations? Note that in the literature, usual period lengths are 10yr calibration + 10yr (different, non-overlapping) validation; maybe the shortest that I could remember is 1yr calibration 2yr validation. I recommend strongly to the authors to use a longer dataset, and split it for calibration and validation.

Klemeš, V. (1986). Operational testing of hydrological simulation models. Hydrological Sciences Journal, 31(1), 13-24.

(2) There is no presentation or discussion of the results in the text. The presentation of the results is limited to give a reference to Table 3, and Figures 6 and 7. One cannot expect a 15-lines section to include "Results and discussion". In the revised paper, the authors need to present quantitatively the performances, and comment (maybe theorize) about why do the methods perform differently? maybe related to routing/travel times? Also, it seems to me that the (exceptionally) high performance of the model is an artifact of the high number of rules used in the FIS, i.e. over-fitting (99 rules is of the same order of magnitude of the number of data points used in the study 12x12=144). The authors need to present alternatives with a smaller set of rules and investigate the decrease of performance. Also, section 3 is titled "Results and discussion", but there is no discussion. The results need then to be discussed in context of similar studies from the literature.

(3) There is no description what-so-ever of the method used. From Page3 Line9 to Page4 Line4 is only textbook knowledge about fuzzy sets absolutely not relevant for

this paper, the can be condensed in 3 sentences. The last paragraph should explain more in detail the Fuzzy Inference System, how the rules work, how they are defined, all in the context of this paper.

(4) In order to make provide some useful take-home message to a potential reader, if would be extremely useful a comparison to alternative methods. This does not need to be extreme long or complex, the authors could for example perform a regression (linear and/or non-linear) with the same predictors as inputs used in the FIS, and compare the performances of both models IN VALIDATION, i.e. in a period different to the one on which the models were calibrated.

(5) There are no conclusions. P6 L18 to P7 L2 is a summary (i.e., another abstract) and P7 L3-10 is a very brief and shallow outlook. There are no conclusions or take-home messages. In this sense, it also not clear what is the novelty of the paper.

(6) There are no references to the relevant river network structures of the region. Looking jointly at Figures 2 and 3, there must be a very important tributary to the Maritza between Svilengrad and Kirişhane, as the discharge difference is large compared to the river reach length (at least I assume it, given that there is no scale in Fig. 2). Also, a brief discussion of the travel times [see comment (2)] would be needed to justify the inputs used in the FIS. Also, a hydrological description of the region is given in P5 L5-10 but, is this related to what happened between 9-21 Feb.2010 in terms of processes [see also comment (1) above] or in terms of flood waves travel times. Related to point (4), it would be useful to compare to results of data-based methods to very simple hydrodynamical (here simple routing, e.g. kinematic or diffusive wave would be enough). Of course, ideally, a comprehensive comparison of the FIS or the regression suggested in (4) could include a forecast rainfall-runoff model, but I understand if this is out of the scope of the present publication.

Please find below a (non-comprehensive) list of other issues that need to addressed in the resubmission:

P1 L7-9: The first 3 sentences of the abstract convey the same message, please condense in one sentence.

P1 L10: What is "short"? Please report length (e.g. in km)

P1 L10: "there is not adequate warning time for Turkey to alert the population against flash floods" I strongly disagree with the formulation of this sentence. It somehow transmits the message that "it is not adequate to alert the population against flash floods", which is disturbing. Please combine with the following sentence, just saying that discharge/water levels from Bulgaria are needed.

P1 L14: "Flow data from the Plovdiv, Svilengrad and Kirişhane stations were gauged every two hours covering the period [...]" I would assume that the discharge data was gauged (measured) for a longer period. I would rephrase as "Discharge data from ... was used with a 2-hours resolution [...]"

P1 L15: I would add something like "The aim is to predict discharge for Maritza@Kirişhane (if this is true) at a given time t. The four models differ "

P1 L20: determination coefficient -> coefficient OF determination P1 L20: Nash-Sutcliffe sufficiency -> Nash-Sutcliffe EFFICIENCY P1 L21: "correlation coefficient": which one? Linear correlation (i.e. Pearson) correlation coefficient?

P1 L22: "[...] models produced highly satisfactory results." - Please be quantitative, and avoid adjectives like "highly satisfactory" (in comparison to what?). Same for Line 24 ("satisfactory"). Please report the most relevant performance values in lines 22-24.

P2 L20: "[...] are capable of more accurate prediction [...]", more accurate than what?

P2 L24 to P3 L6: Please move this to the Data (catchment description) section.

P4 eq(4): The formula of the Coefficient of Determination is wrong. Equation (4) estimates the square of the linear (Pearson, product) correlation coefficient, or $r^2$ (small "r" squared). Capital "R" squared, $R^2$, i.e. the Coefficient of Determination is defined

as 1 - SumSquaresResidual/SumSquaresTotal. In this context this is equivalent to the definition of the Nash-Sutcliffe efficiency. Please note that r2 and R2 ONLY coincide in the case of a simple linear regression. One way to see that eq(4) is not correct is that the Coefficient of Determination may be negative (if the model performance is very poor), and this is impossible with the expression in eq(4).

P5 L12: again the discharge was gauged for a longer period, the authors USED only the mentioned period

P5 L24 "Mamdani model", please give reference

P5 L29 to P6 L1: "After determination of the degrees of the data in different regions in which numbers can be different, the fuzzy rules of the models were generated by using AND logical conjunction (Fig. 4)." I am not sure Figure 4 shows what the authors are describing in this sentence.

P6 L2: Figure 5, in it's current status in not needed

P6 L10 "A summary of the results of the performance criteria of the models is given in Table 3". Please comment the actual results in the section. Also, the name of the section is Results and Discussion and there is no discussion.

P7 L6-9: "Furthermore, long-term observation of the hydro-meteorological conditions of the floods is needed. Hence, developed models are crucial in the prediction of current flows. Doing nothing is not an option; thus, this study has shown that developed fuzzy rule-based models can satisfactorily predict the flow regime with high accuracy and therefore can be used to develop a reliable flow forecasting system." The authors here link the hydro-meteorological processes generating floods with the fuzzy forecast model presented. The data-based method discussed in this paper does not have any base what so ever on processes, it is just a blind fit of the FIS to a single flood event.

P12 Table3: Correlation values of 7.XX are wrong. They are limited, by definition in eq(7) to values between -1 and +1.

---

## Author Comment (AC1) · 8 Sep 2016

All responses to two referees were given in a table with three columns (first column for referee 1, second column for referee 2, and third column for responses) due to existence of similar comments of two referees. Similar comments of referees reviewed in same line of table. In addition, final revised manuscript and a manuscript with highlighted revisions were prepared in Microsoft Word. In the manuscript with highlighted revisions, removed sentences from pre-revised version were depicted as "...removed sentences..." and newly added sentences were depicted as "...newly added sentences...".

[Figure]

Please also note the supplement to this comment:
http://www.nat-hazards-earth-syst-sci-discuss.net/nhess-2016-86/nhess-2016-86-AC1-supplement.zip